# Maximization of the Minicircle DNA Vaccine Production Expressing SARS-CoV-2 RBD

**DOI:** 10.3390/biomedicines10050990

**Published:** 2022-04-25

**Authors:** Cathy Ventura, Dalinda Eusébio, Ana M. Gonçalves, Jorge Barroca-Ferreira, Diana Costa, Zhengrong Cui, Luís A. Passarinha, Ângela Sousa

**Affiliations:** 1CICS-UBI-Health Science Research Centre, University of Beira Interior, Av. Infante D. Henrique, 6200-506 Covilha, Portugal; cathy.ventura@ubi.pt (C.V.); dalinda.eusebio@ubi.pt (D.E.); ggmargarida@gmail.com (A.M.G.); jorgedanielferreira@gmail.com (J.B.-F.); dcosta@fcsaude.ubi.pt (D.C.); 2Associate Laboratory i4HB-Institute for Health and Bioeconomy, Faculdade de Ciências e Tecnologia, Universidade NOVA de Lisboa, 2819-516 Caparica, Portugal; 3UCIBIO-Applied Molecular Biosciences Unit, Departamento de Química, Faculdade de Ciências e Tecnologia, Universidade NOVA de Lisboa, 2829-516 Caparica, Portugal; 4Division of Molecular Pharmaceutics and Drug Delivery, College of Pharmacy, The University of Texas at Austin, Austin, TX 78712, USA; zhengrong.cui@austin.utexas.edu; 5Laboratório de Fármaco-Toxicologia-UBIMedical, Universidade da Beira Interior, 6200-284 Covilha, Portugal

**Keywords:** bioreactor, COVID-19, design of experiments, minicircle DNA vaccine

## Abstract

Nucleic acid vaccines have been proven to be a revolutionary technology to induce an efficient, safe and rapid response against pandemics, like the coronavirus disease (COVID-19). Minicircle DNA (mcDNA) is an innovative vector more stable than messenger RNA and more efficient in cell transfection and transgene expression than conventional plasmid DNA. This work describes the construction of a parental plasmid (PP) vector encoding the receptor-binding domain (RBD) of the S protein from severe acute respiratory syndrome coronavirus 2 (SARS-CoV-2), and the use of the Design of Experiments (DoE) to optimize PP recombination into mcDNA vector in an orbital shaker. First, the results revealed that host cells should be grown at 42 °C and the Terrific Broth (TB) medium should be replaced by Luria Broth (LB) medium containing 0.01% L-arabinose for the induction step. The antibiotic concentration, the induction time, and the induction temperature were used as DoE inputs to maximize the % of recombined mcDNA. The quadratic model was statistically significant (*p*-value < 0.05) and presented a non-significant lack of fit (*p*-value > 0.05) with a suitable coefficient of determination. The optimal point was validated using 1 h of induction, at 30 °C, without the presence of antibiotics, obtaining 93.87% of recombined mcDNA. Based on these conditions, the production of mcDNA was then maximized in a mini-bioreactor platform. The most favorable condition obtained in the bioreactor was obtained by applying 60% pO_2_ in the fermentation step during 5 h and 30% pO_2_ in the induction step, with 0.01% L-arabinose throughout 5 h. The yield of mcDNA-RBD was increased to a concentration of 1.15 g/L, when compared to the orbital shaker studies (16.48 mg/L). These data revealed that the bioreactor application strongly incremented the host biomass yield and simultaneously improved the recombination levels of PP into mcDNA. Altogether, these results contributed to improving mcDNA-RBD biosynthesis to make the scale-up of mcDNA manufacture simpler, cost-effective, and attractive for the biotechnology industry.

## 1. Introduction

Coronavirus disease (COVID-19) is caused by the novel severe acute respiratory syndrome coronavirus 2 (SARS-CoV-2). SARS-CoV-2 is an enveloped, linear single-stranded positive-sense ~30 kb RNA virus, which primarily infects a person through the respiratory tract [1]. This virus is constituted by structural proteins, which are involved in the innate and adaptive responses of the immune system by the host. The virus binds to the angiotensin-converting enzyme 2 (ACE2) receptor on the epithelial cells lining the respiratory tract via its surface spike (S) protein, through the receptor-binding domain (RBD) [2].

Nucleic acid vaccines have been proven to be a revolutionary technology to give a rapid response against pandemics. The first approved vaccine against COVID-19 is based on the messenger RNA-1273 (mRNA) to express S protein and showed for the first time the efficacy, safety and ease of manufacturing of nucleic acid vaccines [3]. Furthermore, ZyCoV-D is the first DNA vaccine approved in India for human use, comprising the plasmid DNA vector pVAX1 encoding the S glycoprotein and the sequence for the IgE signal peptide [4]. In fact, DNA vaccines offer the same advantages of mRNA vaccines, but are more stable, cost-effective and easier to produce and can encode more than one antigen, providing a more complete immune response induced by a multigenic vaccine [5]. In addition, the humoral and cellular immune responses induced by a DNA vaccine can be further improved by altering the route of administration and using special delivery systems and adjuvants [6]. 

Minicircle DNA (mcDNA) is a novel non-viral DNA vector that has become a promising option for gene therapy, DNA vaccines, or intermediate in cell-based therapies [7,8]. This molecule is obtained from the recombination process of the parental plasmid (PP), during the production step inside a specific host, which results in the excision of bacterial gene sequences [9]. The sequences, plasmid replication origin (ORI) and antibiotic resistance gene, are useful for PP cloning and amplification in the bacteria and for selectivity during the host growth [10]. However, the presence of those sequences in the DNA vector delivered to a patient can potentially trigger adverse immune responses and the antibiotic resistance gene can be potentially transferred into human microflora. Although the production of mcDNA is not fully efficient and a small amount of PP may be present at the end of the process, a purification step can be applied to eliminate the PP impurities and further apply the mcDNA vector in therapeutic applications. Additionally, the advantage of using mcDNA instead of pDNA is that the lack of bacterial sequences in the backbone makes the mcDNA in a small DNA vector, which can facilitate its diffusion and increase the biological activity and presents a low immunogenicity [11,12,13]. The mcDNA can express 10- to 1000-fold higher therapeutic transgene products than PP, because the silencing of target gene expression can occur in the PP vector by the formation of repressive heterochromatin around the bacterial sequences [10]. 

The mcDNA biosynthesis can happen within a genetically modified strain, such as *Escherichia coli* (*E. coli*) ZYCY10P3S2T, which allows the in vivo recombination of PP into miniplasmid (mP) (carrying prokaryotic backbone sequences) and mcDNA (carrying therapeutic expression cassette) molecules, by the action of an inducer. This process is controlled by an arabinose inducible pBAD/AraC expression system [5,7,14]. The original PP vector contains the transgene expression cassette flanked with attB and attP genes that are recognized by inducible enzymes and the I-SceI endonuclease recognition site. The presence of L-arabinose will induce the simultaneous expression of PhiC31 integrase (which recognizes attB and attP sequences in PP molecule and induces its recombination into mP and mcDNA) and I-SceI endonuclease (which is responsible for mP and PP degradation due to the recognition of the I-SceI restriction site) by *E. coli* ZYCY10P3S2T [13,15]. Despite the high potential of this technology, the in vivo recombination process is not fully efficient, resulting in samples with low yields of mcDNA and contaminated with its PP precursor [10,15]. 

The design of experiments (DoE) is an analysis tool that can be applied to optimize procedures in a fast and easy manner [16,17]. DoE allows one to define parameters and simultaneously combine them to generate experimental conditions, allowing one to use as few experiments as possible, applying a statistical model [18,19]. Depending on the aim of the work and the number of inputs, several models are available to explore in DoE. The central composite face design (CCF) is a model composed of full or fractional factorial design. This model provides high-quality results since it only uses points within predetermined ranges [20]. 

This work aims to prepare a mcDNA vaccine against COVID-19, optimize the PP in vivo recombination process on a small scale (using orbital shakers) and explore the CCF model, and finally maximize the mcDNA production in a mini-bioreactor platform. The chosen DoE model allows the systematic variation of three inputs simultaneously (antibiotic concentration, induction time, and temperature) to optimize the defined output (% recombined mcDNA). After the prediction and validation of the optimal point to improve the mcDNA yield, the defined recombination conditions were extrapolated for bioreactor experiments to scale up the mcDNA biosynthesis.

## 2. Materials and Methods

### 2.1. Materials 

Terrific Broth (TB) powder and tryptone were from Biokar Diagnostics (Allone, France). Yeast extract was from Biolife (Milano, Italy), glycerol was from Labchem (Zelienople, PA, USA), and L-arabinose was from Alfa Aesar (Tewksbury, MA, USA). The Luria Broth (LB) powder and kanamycin sulfate were from Fisher BioReagents (Pittsburgh, PA, USA). 

### 2.2. Construction of mcDNA-RBD Vector

The pcDNA3-SARS-CoV-2-S-RBD-8his plasmid was a gift from Dr. Erik Procko (Addgene, MA, USA-Plasmid #145145) and was used as template to amplify the secreted receptor-binding domain (RBD; a.a. 333-529) of S protein from SARS-CoV-2 (NCBI Reference Sequence: NC_045512.2 (21563..25384)) [21]. The pMC.CMV-MCS-EF1-GFP-SV40Poly A plasmid (System Biosciences, CA, USA) was used as the starting plasmid for cloning the RBD gene sequence. 

The GRS Taq DNA polymerase (GRiSP, Lisbon, Portugal) was used to perform polymerase chain reaction (PCR). The components of the mix for each PCR reaction were 0.75 µL MgCl_2_ (25 mM), 0.25 µL dNTPs (10 mM each), 1.25 µL PCR buffer, 100 ng template DNA, 0.25 µL Taq DNA polymerase (5 U/µL) and PCR-grade water up to 12.5 µL. Specific primers were used with XbaI and BamHI restriction sites, with a final concentration of 0.16 µM (FW: 5′- AAT CTA GAA TGA AGA CCA TCA TCG CCC T -3′; RV: 5′- ATG GAT CCT CAA TGA TGG TGG TGG T -3′). After an initial cycle of 5 min at 95 °C, 30 cycles were performed for 30 s at 95 °C, 30 s at 60 °C, and 1 min at 72 °C for extension. After amplification, a final extension step was included of 10 min at 72 °C. The PCR products were analyzed by 1% agarose gel electrophoresis. The GRS Ladder 1kb (GRiSP, Porto, Portugal) was used as DNA molecular weight marker.

The XbaI (Takara Bio, CA, USA) and BamHI (NZYTech, Lisbon, Portugal) restriction enzymes were used for the enzymatic digestions of the starting plasmid and RBD insert, according to the manufacturer’s instructions. Between digestions, the starting plasmid was incubated with alkaline calf intestinal phosphatase (CIP) (New England Biolabs, MA, USA), according to manufacturer’s instructions. The GRS PCR and Gel Band Purification Kit (GRiSP, Porto, Portugal) was used to purify DNA fragments from PCR reactions and enzymatic restriction digestions. To perform ligation reactions, the T4 DNA ligase enzyme (NZYTech, Lisbon, Portugal) was used. Following ligation protocol, 1:3, 1:6, and 1:9 molar ratios of vector:insert were tested and 50 ng of the digested vector was used. As recommended for cohesive-end ligations, both vector and insert DNA were heated prior to the ligation for 10 min at 56 °C. Reaction mixtures were incubated for 16 h at 18 °C and afterward these mixtures were used to transform competent *E. coli* TOP10 cells. Positive colonies were identified by PCR screening test, and one transformed colony was placed to grow in a liquid medium. Afterward, the cloned pDNA was purified with a NZYMiniprep kit (NZYTech, Lisbon, Portugal) and used to transform competent *E. coli* ZYCY10P3S2T cells. After the identification of positive colonies by a PCR screening test and their growth in liquid medium for plasmid purification, the success of the cloning procedure was confirmed by DNA sequencing.

### 2.3. Fermentation and Induction Conditions in an Orbital Shaker

First, *E. coli* ZYCY10P3S2T cells transformed with the PP-RBD (7.1 kbp) vector were inoculated in LB-agar plates supplemented with kanamycin (50 µg/mL) and incubated overnight at 37 °C. Subsequently, some colonies were transferred to a 250 mL Erlenmeyer containing 62.5 mL of TB medium (20 g/L of tryptone; 24 g/L of yeast extract; 4 mL/L of glycerol; 0.017 M KH_2_PO_4_, 0.072 M K_2_HPO_4_, pH 7.0). To evaluate the effect of different growth conditions on the amplification of PP-RBD, two fermentation temperatures were tested. Thus, each pre-fermentation Erlenmeyer was placed in an orbital shaker at 37 °C or 42 °C, with constant stirring (250 rpm), until the optical density at 600 nm (OD_600nm_) reached ~2.6. Once the growth exponential phase was achieved, a certain volume was transferred to another 250 mL Erlenmeyer containing 62.5 mL of TB medium, so that fermentation started with an OD_600nm_ of 0.2, avoiding the adaptation step in the fermentation procedure. Following this, the Erlenmeyer was placed in the orbital shaker at 37 °C or 42 °C and with constant stirring (250 rpm) until an OD_600nm_ of 5 was reached. 

To evaluate if TB medium affects mcDNA-RBD recombination, two different strategies were investigated. In the first strategy, the fermentation medium containing cells was directly transferred/mixed to the induction medium, and in the second strategy, the cells were centrifuged to remove the TB medium and then transferred to the induction medium. For the samples which were subjected to centrifugation, 2000× *g* rpm for 20 min at room temperature was applied to avoid any damage or stress to the cells. Subsequently, the pellet was resuspended in 5 mL of LB medium and transferred into a 500 mL Erlenmeyer with 120 mL of LB medium, containing 0.04 M NaOH to ensure pH 7.0 and L-arabinose 0.01% (*w*/*v*) (Figure 1). For both strategies, the induction process was carried out for 2 h at 32 °C with constant stirring at 250 rpm, as described by Gaspar and co-workers [12]. After fermentation or induction, 10 mL of cells were recovered by centrifugation at 3900 rpm for 10 min at 4 °C and stored at −20 °C. The different samples were extracted using a GeneJET Plasmid Miniprep Kit (ThermoFisher, MA, USA) following the supplier’s protocol and total DNA concentration was quantified through a NanoPhotometer^TM^. Results were analyzed by 1% agarose gel electrophoresis.

### 2.4. Design of Experiments (DoE)

DoE was used to optimize the PP recombination into mcDNA through the application of the CCF model. For this purpose, the inputs A, B and C (which correspond to the induction temperature (°C), antibiotic concentration (µg/mL), and induction time (h), respectively) were selected, while the % of recombined mcDNA was the output to be maximized. The inputs were studied at three levels (−1; 0; +1). The software Design-Expert version 11 was applied the CCF model and proposed a total of 17 experiments, considering three replicates of the central point. After performing all the experiments and assessing the respective outputs, the results were included in the DoE software and all statistical analysis were performed to validate the model suitability applied in DoE. The generalized second-order polynomial model equation used is presented below (Equation (1)):(1)$$Y=\beta_{0}+\beta_{1}X_{1}+\beta_{2}X_{2}+\beta_{3}X_{3}+\beta_{11}X_{1}^{2}+\beta_{22}X_{2}^{2}+\beta_{33}X_{3}^{2}+\beta_{12}X_{1}X_{2\;}+\beta_{13}X_{1}X_{3}+\beta_{23}X_{2}X_{3}\;$$

The optimal point to maximize PP recombination into mcDNA was predicted by the DoE and validated by performing three assays as replicates of proposed conditions. Finally, the average in % of recombined mcDNA obtained in these replicates should be within the confidence interval (CI) of 95% provided by the DoE for the chosen optimal point.

### 2.5. Agarose Gel Electrophoresis

An agarose gel with a concentration of 1% (*w*/*v*) was prepared in TAE buffer (40 mM Tris base, 20 mM acetic acid, 1 mM EDTA, pH 8.0) and stained with 0.012 μL/mL of GreenSafe premium (NZYTech, Lisbon, Portugal) to analyze PP and mcDNA molecules. Electrophoresis was run at 150 V for 40 min and analyzed under ultraviolet (UV) light using a FireReader Imaging System (UVITEC, Cambridge, UK). To minimize effects that could compromise DoE results, the experimental conditions were divided by temperature subgroups. Therefore, after each miniprep extraction, nucleic acids were run on individual agarose gels under the same conditions.

### 2.6. Output Determination

The optimization of PP recombination into mcDNA was analyzed through the ratio between the relative density of the band corresponding to mcDNA and the relative density of the band corresponding to PP, using the software Image Lab, version 6.1. 

### 2.7. Analytical Chromatography

The quantitative assessment of the total pDNA was done with the CIMac^TM^ pDNA analytical column, as described previously by our research group [22]. The chromatographic runs were performed in an AKTA Purifier system (GE Healthcare, Buckinghamshire, UK) with UNICORN^TM^ 5.11 software. A calibration curve of the pDNA was constructed with the concentration range of 1–100 µg/mL. Briefly, the samples were prepared by diluting with 200 mM Tris buffer (pH 8.0). Following this, the analytical column was equilibrated with 600 mM NaCl in 200 mM Tris buffer (pH 8.0) and after the injection of 50 µL of each sample, a linear gradient of 10 min to 700 mM NaCl in 200 mM Tris buffer (pH 8.0) was applied. The chromatographic runs were carried out at 1 mL/min at room temperature and the absorbance was continuously monitored at 260 nm. The peaks were integrated to apply in the calibration curve for quantification of the total pDNA present in extracted samples. 

### 2.8. Fermentation and Induction Conditions in Mini-Bioreactor

To initiate *E. coli* ZYCY10P3S2T bioreactor culture, cells were inoculated in LB-agar plates, and a pre-fermentation was performed following the same procedures described above. Subsequently, the fermentation and induction processes were carried out in a 750 mL bench-top parallel mini-bioreactor (Infors HT, Bottmingen, Switzerland) with 250 mL of sterilized TB or LB medium, respectively. The pH was kept constant in both media at 7.0 by the automatic addition of 0.75 M H_2_SO_4_ and 12.5% (*v*/*v*) NH_4_OH through two peristaltic pumps. The dissolved oxygen (DO) was controlled by a three-level cascade of stirring, between 250 and 950 rpm and the mass-flow between 0.2 and 2 vvm. The temperature was set at 42 °C for the fermentation and lowered to 30 °C in the induction step. During the entire process, foaming was manually controlled by the addition of an antifoam agent (Antifoam A, Sigma-Aldrich, St. Louis, MO, USA) at 1% (*v*/*v*) concentration. The parameters and their profiles were maintained under constant control through the IRIS software (Infors HT, Bottmingen, Switzerland). Briefly, the implemented process for the scale-up of biomass yield and PP recombination into *E. coli* ZYCY10P3S2T bioreactor cultures have two main stages—fermentation and induction. In the fermentation process, previously established operational and experimental conditions for Erlenmeyer cultures were maintained, although several oxygen pressure (pO_2_) values were analyzed: 20, 30, and 60%. In turn, throughout the induction phase, the influence of pO_2_ (20, 30, and 60%), L-arabinose inducer concentration (0.01 and 1%), and induction time point (5, 10, and 24 h) were evaluated. The cellular suspensions were harvested by centrifugation at 3900 rpm for 10 min at 4 °C, and then stored at −20 °C until further use.

### 2.9. L-Arabinose Determination by HPLC Analysis

To detect and quantify the consumption of L-arabinose during the induction step in the mini-bioreactor, a calibration curve was constructed, implementing a similar method as described by Pedro and collaborators [23]. The chromatographic analysis was performed using an HPLC model Agilent 1260 (Agilent, Santa Clara, CA, USA) equipped with an autosampler and quaternary pump coupled to a 1260 Infinity Refractive Index Detector (RID) (Agilent, CA, USA). The chromatographic separation was achieved on a cation-exchange analytical column Agilent Hi-Plex H (300× 7.7 mm i.d.; 8 μm), acquired from Specanalítica (Lisbon, Portugal). The analysis was performed at 50 °C with a flow rate of 0.5 mL/min using an isocratic elution with 0.005 M H_2_SO_4_. The mobile phase was filtered prior to the analysis under vacuum using a 0.2-μm pore nylon membrane and degassed for 15 min in an ultrasonic bath. A calibration curve was constructed with L-arabinose standards (0.001–1%), as indicated in Equation (2):(2)$$y=3\times 10^{6}\;x+43,265$$

For the analysis of L-arabinose consumption, the collected samples were then centrifuged at 3900× *g* rpm for 10 min, and the supernatant was recovered and filtered through a 0.22-μm cellulose-acetate filter.

## 3. Results and Discussion

### 3.1. Cloning of RBD Gene Sequence into PP Vector

The presented work started with the construction of a PP vector encoding the RBD of S protein from SARS-CoV-2. First, a PCR amplification was performed with specific primers designed to amplify the RBD insert and using the pcDNA3-SARS-CoV-2-S-RBD-8his plasmid as template, as described in the previous section. The XbaI and BamHI enzymes were then used to recognize the restriction sites present in both RBD insert and PP vector, specifically in the multiple cloning site (MCS), as schematically represented in Figure 2, to acquire the same cohesive ends and facilitate the further ligation step. After the purification of digested products, resultant fragments were analyzed by 0.8% agarose electrophoresis. As shown in Figure 3, both PP vector (7.0 kbp) and RBD insert (640 bp) display good integrity. Furthermore, the PP vector was dephosphorylated between digestions to prevent vector self-ligation during the cloning step. Afterward, DNA T4 ligase enzyme was used to perform the ligation of the RBD insert into the PP vector. Three different molar ratios of vector:insert were tested (1:3, 1:6, 1:10) and *E. coli* TOP10 competent cells were transformed with previous cloning mixtures by heat shock. The isolated colonies that had grown on an LB-agar plate containing kanamycin (50 µg/mL) were used for PCR screening tests to confirm the RBD insert presence and one positive colony was selected to grow in liquid medium. After pDNA purification, the vector was used to transform *E. coli* ZYCY10P3S2T, the minicircle producer strain, and an isolated colony was used for a PCR test to confirm the presence of the RBD insert. The positive colony was cultivated in liquid medium and the purified vector was used for DNA sequencing, to confirm the alignment between RBD insert and cloned PP vector (Figure 4). The results revealed that the RBD insert was successfully cloned into the PP vector. The sequenced colony was used to create cryopreserved bacterial banks to explore the best conditions for PP-RBD production and its recombination into mcDNA-RBD.

### 3.2. PP Amplification and Recombination into mcDNA

Minicircles are small molecules devoid of bacterial sequences, presenting great therapeutic interest due to their low immunogenicity [11]. Moreover, an mcDNA vector exhibits increased transfection efficacy and transgene expression when compared to its PP precursor [15,24]. However, the lack of total recombination of PP into mcDNA during the induction step and the similarity between these biomolecules has a negative impact on the subsequent purification stages. Thus, after PP-RBD vector construction, this work aimed to optimize PP recombination into mcDNA. 

For this purpose, preliminary experiments were performed to establish the best temperature conditions (37 °C or 42 °C) of the fermentation step and its indirect effect on the induction step, as well as the best way to transfer the cells between these two steps (with or without centrifugation), evaluating their influence on both PP and mcDNA levels. After each fermentation or induction step, cells were recovered, PP or mcDNA molecules were extracted, and the results were analyzed by agarose gel electrophoresis. In lanes 1 and 2 of Figure 5 are presented the influence of temperature in PP amplification during the fermentation step. It is clear that the PP integrity was maintained at both fermentation temperatures (lanes 1 and 2, Figure 5). However, the experiment at 42 °C (lane 2) presents a higher band density for PP than the one performed at 37 °C (lane 1). In line with previous studies, different fermentation temperatures can influence the pDNA final yield. For instance, using a genetically modified *E. coli* strain, fermentation at 42 °C increased specific plasmid yield in comparison with the yields obtained at 37 °C [25,26].

Afterward, the indirect influence of temperature on the PP-RBD recombination process in the induction step combined with the cell transference in TB medium (without centrifugation) or without TB medium (after cell centrifugation) was evaluated. Therefore, for both fermentation temperatures (37 °C and 42 °C), two 250 mL Erlenmeyers with 62.5 mL of TB medium followed different paths. In the first one, a total of 62.5 mL of TB medium was added directly to a 500 mL Erlenmeyer with 62.5 mL of LB medium. In the second one, the total 62.5 mL of TB medium was centrifuged and the supernatant was discarded to recover the cells, which were resuspended in 125 mL of LB medium and placed in a 500 mL Erlenmeyer. These results are present in Figure 5 and demonstrate an increase in mcDNA content when the cell centrifugation was performed (lanes 5–73 mg/L and 6–76 mg/L) in comparison to experiments without centrifugation (lanes 3–60 mg/L and 4–64 mg/L). This behavior can be explained due to the glucose present in the TB medium that is transferred to the induction step, which could repress the pBAD/AraC promoter, and consequently, inhibit PP recombination into mcDNA [9]. 

Thus, the best conditions for PP production and cell transference to the induction step should be conducting the fermentation step at 42 °C and removing glucose from the TB medium by cell centrifugation before starting the induction process in LB medium containing L-arabinose. Despite these achievements, the conditions for PP recombination into mcDNA during the induction step need to be explored and optimized. DoE is a suitable tool to quickly accomplish this aim using as few experiments as possible.

### 3.3. DoE Inputs for PP Recombination into mcDNA

Besides the PP amount reached in the fermentation step, several parameters can influence PP recombination and consequently the mcDNA biosynthesis during the induction step, such as inductor concentration, antibiotic concentration, induction time, and induction temperature. However, previous studies performed at a small scale using orbital shakers have explored different L-arabinose concentrations and concluded that an improvement in inductor concentration is not reflected in a mcDNA yield increase [15]. Based on these assumptions, we decided to start this work with 0.01% of L-arabinose, which was the lowest concentration that revealed a very satisfactory mcDNA yield. 

Considering that our goal was to improve mcDNA yields, the other three parameters were explored as the DoE inputs. To define the input ranges, data from previously published works were considered [12,15,27]. The parameters were explored with a defined range of 0–50 µg/mL for kanamycin concentration, given that 50 µg/mL was already used in our research group [12]; 1–5 h for induction time, since it is not recommended to extend the induction step more than 5.5 h in an orbital shaker [27]; and 30 °C–38 °C for the induction temperature, because PhiC31 integrase and I-SceI endonuclease enzymes involved in the recombination process had different optimal activity temperatures—32 °C and 37 °C, respectively [28]. As the output defined was the % of recombined mcDNA and we only intend to explore points within predetermined ranges, the CCF design was chosen for this work. This model proposed 17 experiments to be performed with different input conditions, with three replicates of the central point, as indicated in Table 1. 

To conduct each individual experiment, considering the induction conditions proposed by the DoE, the cells were recovered and the mcDNA was extracted and analyzed by agarose gel electrophoresis, as depicted in Figure 6. Each electrophoresis lane was analyzed using the software Image Lab to evaluate the percentage of recombined mcDNA from each experiment to be included in the DoE as the respective output. This percentage was calculated through the ratio between the relative density of the band corresponding to mcDNA and the relative density of the band corresponding to PP (Table 1). Through the analyses of all results, it seems the most promising conditions to obtain high mcDNA yield during the induction step are 30 °C, without the presence of kanamycin, for 1 h (run 2 of Table 1 and Figure 6), given that 92.75% of recombined mcDNA was obtained. When comparing conditions without kanamycin presence and with a concentration of 50 µg/mL (lanes 1 and 5), a decrease in the mcDNA amount is observed (81.43% to 47.85%). This behavior can be related to some toxicity induced by the antibiotic. As the recombination process advanced, the bacteria lose antibiotic resistance since the selection marker gene is degraded by I-SceI endonuclease upon the PP and mP recognition. Observing experiments without kanamycin, during 1 h of induction, at opposite temperatures (runs 2 and 14), some changes in mcDNA yields indicate that when the temperature increases, 30 °C to 38 °C, the recombined mcDNA decreases, respectively, from 92.75% to 38.84%. The satisfactory results at low temperatures can be explained by the fact that the ΦC31 integrase has optimal activity at low temperatures and the endonuclease I-SceI presents a minimal activity under these conditions [28]. Thus, no degradation of PP will occur before its recombination. When the induction step is prolonged for 5 h, regardless of other established conditions, a gradual decrease in the mcDNA yield is noticed. This evidence indicates that there is an increase in metabolic stress that favors cell lysis and death during a prolonged induction phase, and the recombined mcDNA is degraded through time [27]. 

### 3.4. Model Generation and Statistical Analysis

After the accomplishment of all experiments proposed by the CCF design and assessing the outputs, statistical analysis was performed by Design-Expert software. In Table 2 are the statistical coefficients obtained for the % of recombined mcDNA, which are used to understand if the statistical model generated from these experiments is valid and fits the data. Thus, R^2^ represents the coefficient of determination, providing information regarding the fitness of the output statistical model to the data [17]. This value varies between 0 and 1, with close to 1 being desirable. As is perceivable in Table 2, the R^2^ of the output is 0.9972, suggesting the model fits the data. Adjusted R^2^ represents the theoretical values being adjusted to the experimental data [29]. The output presents a valid adjusted R^2^ since it only decreased by 0.0035 compared to its R^2^. The predicted R^2^ provides information concerning the suitability of the model in predicting new data. The model presents a high predicted R^2^ value (0.9762), thus highlighting the predictive power of this model. At last, adequate precision allows the measurement of the signal-to-noise ratio [16]. This parameter must be higher than four to indicate an adequate signal. According to Table 2, our ratio of 60.885 indicates an excellent signal and suggests this model can be used to navigate the design space. 

Observing all these coefficients, the quadratic model was chosen to proceed with the statistical analysis of this output. To further prove the validity of the DoE, ANOVA analysis was performed. In Table 3 is represented the model significance for the output % of recombined mcDNA, including all the parameters used in this model, coupled with the corresponding lack of fit. A good valid model must present a significant value for the model (*p*-value < 0.05) and a non-significant value for the lack of fit (*p*-value > 0.05), thus suggesting the model data are significant and fit [16]. According to Table 3, all the model values are significant and do not present a significant lack of fit. Consequently, it can be confirmed that a good and valid statistical model was achieved for this output.

### 3.5. Input Effects on % of Recombined mcDNA

To evaluate the main effects that each input presents towards the % of recombined mcDNA, a coded multiple regression equation was generated by Design-Expert software. In this equation, the signal behind each factor indicates a positive or negative effect in the response [30]. In Equation (2) is presented the output regression equation, where A represents induction temperature (°C), B represents antibiotic concentration (µg/mL), and C represents induction time (h). Through the equation, it can be deduced that the induction temperature has a negative effect on the % of recombined mcDNA. As mentioned before, the temperature effect can be related to the temperature at which ΦC31 integrase has optimal activity [28]. Since ΦC31 integrase will be responsible for PP recombination into mP and mcDNA, and displays an optimal activity at low temperatures, the increase in this factor seems to be critical for the percentage of recombined mcDNA [10]. The antibiotic concentration increment also has a negative effect on the output. During the induction phase, the endonuclease I-SceI will degrade the mP and unrecombined PP, which possess the antibiotic resistance gene and recognition sequence for this enzyme. After this digestion, the bacteria lose antibiotic resistance, and the presence of antibiotics in the induction medium will be a stress factor that can cause cell death. Finally, the induction time shows a negative effect which may indicate that the bacteria enter a state of stress and eventually die, and the recombined mcDNA is degraded [27].
% of recombined mcDNA = +636.57 − 25.90 A − 1.54 B − 35.47 C + 0.04 AB + 0.82 AC − 0.04 BC + 0.27 A^2^ + 9.18 × 10 ^−4^ B^2^ − 0.06 C^2^
(3)

### 3.6. Output Optimization and Model Validation

After validation of the statistical model and understanding the effect that each factor has on the output, conditions of each input to reach the optimal point (maximizing the percentage of recombined mcDNA) were predicted. Design-Expert software suggested the combination of the induction temperature at 30 °C, the absence of antibiotics, and 1 h of induction time, to obtain a % of recombined mcDNA between 88.72% and 95.50% of the confidence interval for the validation of the optimal point. These conditions were applied in three independent experiments and the average of the resulting outputs provided 93.87 ± 3.24 % of recombined mcDNA. This value is within the confidence interval provided by the Design-Expert software, where the output is considered valid, according to Table 4. 

Overall, through the use of experimental design, the optimization of PP recombination process into mcDNA was successfully achieved. The induction temperature, antibiotic concentration and induction time revealed important factors that strongly influence the output. 

### 3.7. Determination of PP and mcDNA Concentration

After optimizing the PP fermentation and establishing the optimal point of recombined mcDNA, the quantitative analysis of the total pDNA amount present in each sample was determined with the CIMacTM pDNA analytical column, as previously described [22]. Thus, the PP concentration reached after the fermentation process optimization in an orbital shaker was 23.07 mg/L. Considering that large amounts of pDNA are required for biopharmaceutical applications, several studies have attempted to improve the pDNA production in shake flasks. For example, Galindo and co-workers used an enzyme-controlled glucose release system and the *E. coli* DH5α strain that resulted in a pDNA production of 26.6 mg/L in shake flasks [31]. In another study, a proteome-reduced *E. coli* strain produced 74.8 mg/L of pDNA when cultured in a fed-batch mode in shake flasks [32]. In addition, Kay and collaborators used the genetically modified *E. coli* ZYCY10P3S2T strain that expresses a set of inducible minicircle-assembly enzymes. These authors were able to obtain PP concentrations between 5.45 and 5.84 mg/L using TB medium containing kanamycin (50 µg/mL) at 37 °C [15]. In fact, the variability of these results can be strongly dependent on the *E. coli* strain used, which can influence and limit the pDNA production process. Given that we are using the same strain as Kay and coworkers, our production conditions, especially the temperature of 42 °C, allowed a considerably superior PP concentration. 

Regarding the recombined mcDNA obtained in this work with the optimal conditions, a concentration of 16.48 mg/L was achieved. In this particular case, Kay and collaborators were able to obtain mcDNA yields between 3.40–4.83 mg/L, mixing the fermentation medium and the minicircle induction mix in the same proportion (1:1) and incubating at 32 °C for 5 h [15]. Remarkably, our process optimization performed with DoE allowed us to significantly improve the mcDNA yield, obtaining one almost 20 times higher than those reported in the literature, by recovering cells through centrifugation before the start of the induction process to eliminate glucose and the presence of kanamycin, decreasing the temperature from 32 °C to 30 °C, and only requiring 1 h of induction. 

### 3.8. Scale up of the mcDNA Biosynthesis in Mini-Bioreactor

As previously mentioned, the bioreactor cultures of *E. coli* ZYCY10P3S2T were divided into two main phases, fermentation and induction. The fermentation phase has the main goal of increasing PP yield associated with biomass production in these cultures. Therefore, we applied identical operational and environmental conditions as for orbital shaker cultures. In addition, we investigated the impact of different levels of dissolved oxygen on cell growth that could maximize PP production in this initial stage. Considering culture conditions explored by other research groups for pDNA fermentation in bioreactors, we choose 30% pO_2_ as starting point for the scale-up of our PP production [33,34,35]. In this preliminary assay, bacterial samples were collected every hour to monitor OD_600nm_ evolution, and the result revealed that cells attained a stationary phase after 6 h of fermentation (OD_600nm_ = 27.7 ± 0.1). Subsequently, a bioreactor culture using 60% pO_2_ was performed and after 6 h of fermentation, the cells reached an OD_600nm_ = 42.0 ± 0.8. Thus, the bacterial amplification using 60% pO_2_ resulted in higher cell levels than those obtained at lower oxygen pressure. Afterward, the bacterial growth of *E. coli* ZYCY10P3S2T using 60% pO_2_ was characterized to determine the evolution profile during fermentation, being observed in a stationary phase between 5 and 6 h. Considering that the induction phase should be performed at the end of the exponential growth phase, in the following bioreactor assays, the induction was performed after 5 h of fermentation. Additionally, the analysis of agarose gel (data not shown) by densitometry revealed that the production of PP was enhanced by a factor of 37.22 and 65.46 for 30 and 60% pO_2_, respectively, when compared to orbital shaker experiments. In line with previous studies, higher plasmid productivity and faster cell growth were obtained at higher dissolved oxygen concentrations [36,37]. 

Since the biomass obtained in the bioreactor is eight times higher than the biomass obtained in the orbital shaker (OD_600nm_ of 42 and 5, respectively), the influence of the L-arabinose concentration (0.01 and 1%) was explored in bioreactor induction step to examine whether some improvement can be obtained. Thus, the optimized conditions for the recombination of PP into mcDNA obtained in the orbital shaker were the first ones applied to the bioreactor induction step (temperature of 30 °C, without antibiotic and 0.01% of L-arabinose), applying 30% pO_2_. The same assay was performed, although with the L-arabinose concentration changed to 1%. The PP recombination into mcDNA was incremented by a factor of 60.86 and 61.19, respectively. In parallel, the L-arabinose consumption from the culture medium during these induction steps in the bioreactor was assessed by HPLC coupled with RID, using a cation-exchange analytical column Agilent Hi-Plex H [22]. For this, a calibration curve with several L-arabinose standards (range from 0.001 to 1%) was constructed, as previously indicated in Equation (2). Results presented in Table 5 demonstrated the method’s capacity to detect L-arabinose as well as its consumption is more evident during after 4 or 6 h of induction. Nevertheless, despite the amount of L-arabinose consumed in the medium, the mcDNA increment was minimal. Thus, an additional supplement of the L-arabinose inducer during the induction phase is not required [38] and an L-arabinose concentration of 0.01% was used in the following assays. On the other hand, as the scale-up provides an increase in biomass, the induction time may not be sufficient for the maximum recombination of PP into mcDNA to occur. Therefore, to evaluate the time influence on the recombination process, 5, 10, and 24 h of induction were explored at 30% of pO_2_. However, no significant differences were found in the recombination rate after 5 h of induction, and the enhancement of the amount of mcDNA was a factor of 69.97 when compared to orbital shaker, and after that time point, the amount of PP and mcDNA started to decline, suggesting that 5 h is enough to maximize the recombination process under the operating conditions tested.

The last parameter that was explored in the bioreactor induction step was the pO_2_ influence, being studied at 30 and 60%. The results showed that PP recombination into mcDNA was incremented by a factor of 69.97 and 43.67, respectively, when compared to orbital shaker studies. This result suggests that the acceleration of the biomass growth/amplification at 60% pO_2_ did not favor the induction step, as occurred in the fermentation step. Probably, during the induction step, the increase in cellular metabolism interferes with the time that cells need for the expression of ΦC31 integrase and endonuclease I-SceI and their action in PP recombination and miniplasmid degradation [27]. In addition, these conditions will ensure the plasmids’ structural and segregation stability during the induction step. Thus, the most favorable conditions for each phase of the bioreactor cultures were 60% pO_2_ for fermentation for 5 h and 30% pO_2_ for induction, with 0.01% L-arabinose for recombination for 5 h. 

As described previously, we determined the concentration of PP and mcDNA obtained in the orbital shaker. For the bioreactor, we estimated that the PP concentration was 1.51 g/L (Table 6) at the end of the fermentation step. Other studies in bioreactors have demonstrated pDNA yields between 0.18–218 mg/L [35,39] and 26.59–229.8 mg/L [33], which indicate that the bioreactor bioprocess implemented here could be used to increase the PP yield using the *E. coli* ZYCY10P3S2T host and consequently will be a good starting point to improve the recombination levels of PP into mcDNA. For the mcDNA, we estimated a concentration of 1.15 g/L (Table 6). Concerning the only study, to our knowledge, about the yield of mcDNA attained in a mini-bioreactor platform, Šimčíková and colleagues developed a strategy to improve mcDNA yields by optimizing the parA gene 5′ untranslated region of the BW2P *E. coli* strain, achieving recombination efficiency of approximately 80% and a total plasmid concentration (PP, mcDNA, and mP) of 50 mg/L [40]. However, these authors used different fermentation conditions, LB medium, and a temperature of 37 °C, while in the present work, TB was used with a fermentation temperature of 42 °C. These differences can influence the obtained initial biomass, which may explain the increase in our mcDNA yield. 

## 4. Conclusions

COVID-19 remains a global threat, and although the most critical phase of the spread of this disease and related deaths appears to be under control, the development of new vaccines will continue to be a global need. In fact, DNA vaccines have already been revealed to have great potential against this pandemic. In this work, several strategies were explored in order to optimize PP recombination into the cutting-edge mcDNA vector, encoding the RBD sequence from SARS-CoV-2 S protein, and to scale-up its biosynthesis. Therefore, the optimal fermentation temperature to obtain the highest yield of PP-RBD was determined to be 42 °C. The PP recombination into mcDNA was also optimized by resorting to a DoE tool, the induction temperature being defined as 30 °C for 1 h without antibiotic presence, and performing cell centrifugation at the end of fermentation to completely remove the presence of glucose before starting the induction step. This approach allowed obtaining yields of 93.87% of mcDNA-RBD in an orbital shaker, which corresponds to a concentration of 16.48 mg/L. To scale up the mcDNA biosynthesis in a mini-bioreactor, other parameters were explored in fermentation and induction steps, such as the pO_2_. Thus, applying 60% pO_2_ during 5 h of fermentation, PP was enhanced by a factor of 65.46 (corresponding to a concentration of 1.51 g/L) and using 30% pO_2_ and 0.01% L-arabinose, without antibiotic, for 5 h of induction, mcDNA was increased by a factor of 69.97 (corresponding to a concentration of 1.15 g/L) when compared to the orbital shaker experiments. Altogether, these results improved mcDNA biosynthesis and explored its scale-up, and are a strong contribution to the simpler, more cost-effective, and attractive manufacture of this new and promising biopharmaceutical for the biotechnology/biopharmaceutical industry.

## Figures and Tables

**Figure 1 biomedicines-10-00990-f001:**
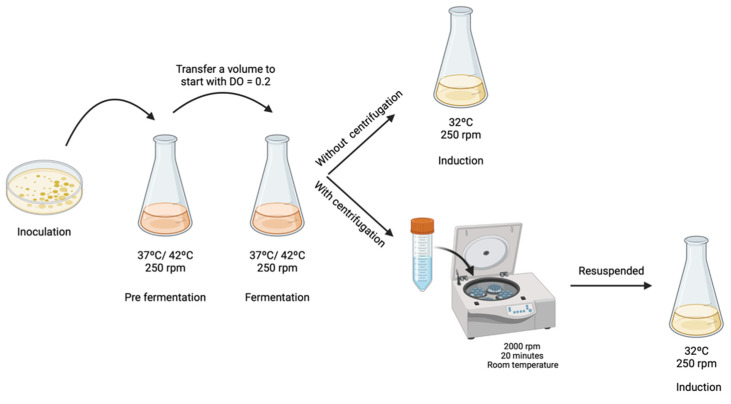
Schematic representation of different procedures to start the induction step.

**Figure 2 biomedicines-10-00990-f002:**
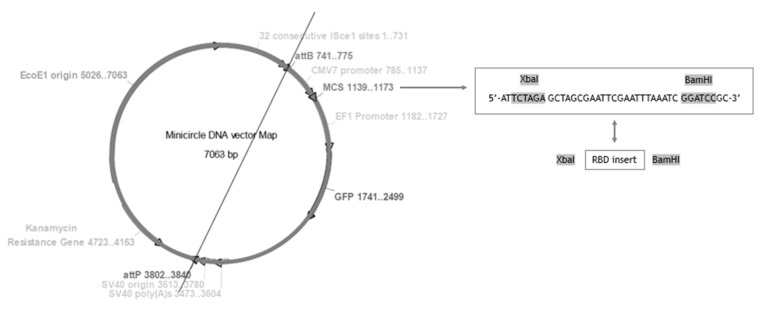
Minicircle DNA vector map for PP-RBD construction and restriction enzyme map.

**Figure 3 biomedicines-10-00990-f003:**
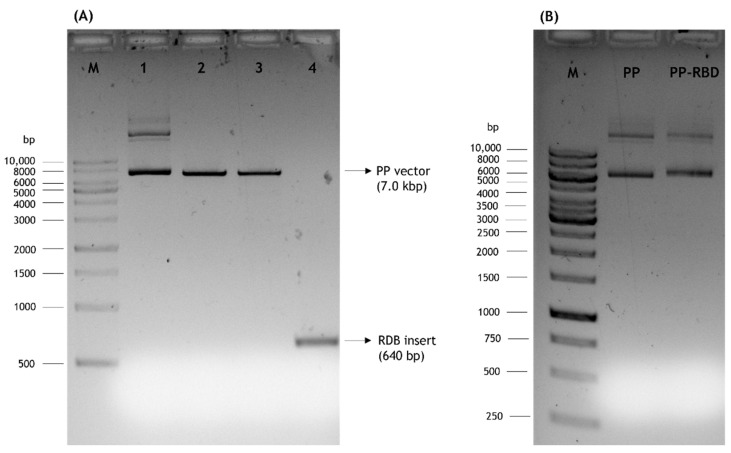
Analysis of DNA fragments by agarose gel electrophoresis. M–molecular weight marker. (**A**) 1–PP vector; 2–PP vector digested with XbaI; 3–PP vector digested with XbaI and BamHI; 4–RBD insert digested with XbaI and BamHI. (**B**) Initial PP and PP-RBD vector.

**Figure 4 biomedicines-10-00990-f004:**
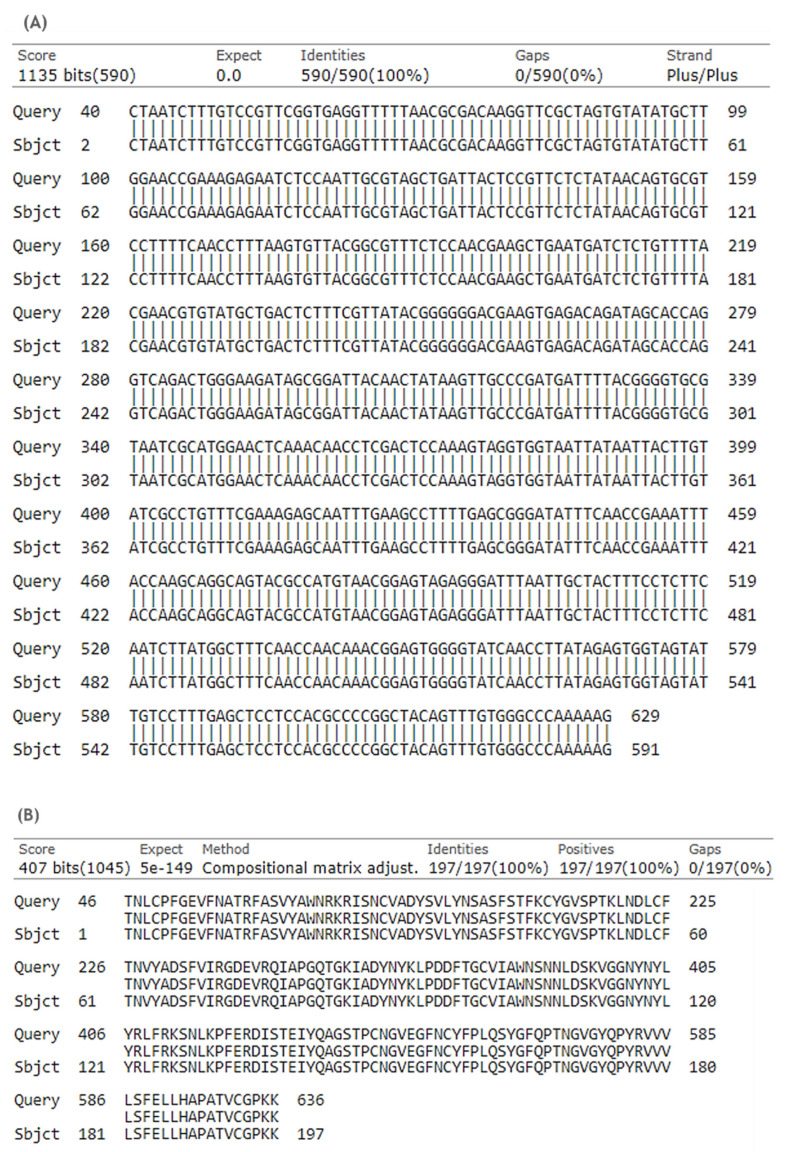
Alignments of RBD-cloned PP vector and RBD sequence performed with basic local alignment search tool (BLAST). (**A**) Nucleotide alignment. (**B**) Amino acid alignment. The query corresponds to PP-RBD vector sequence and the subject is the sequence corresponding to RBD (a.a. 333-529) of S protein from SARS-CoV-2 (NCBI reference sequence: NC_045512.2 (21563..25384)).

**Figure 5 biomedicines-10-00990-f005:**
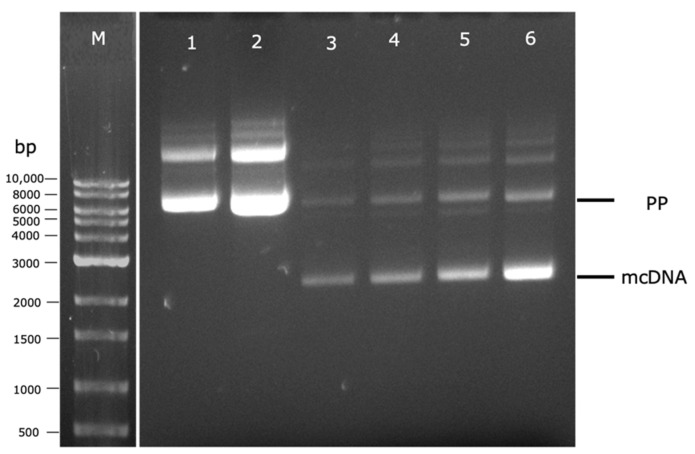
Analysis of PP and mcDNA content, after fermentation or induction steps, by agarose gel electrophoresis. M–molecular weight marker. (1) Fermentation at 37 °C; (2) Fermentation at 42 °C; (3) Fermentation at 37 °C and induction without centrifugation; (4) Fermentation at 42 °C and induction without centrifugation; (5) Fermentation at 37 °C and induction with centrifugation; (6) Fermentation at 42 °C and induction with centrifugation.

**Figure 6 biomedicines-10-00990-f006:**
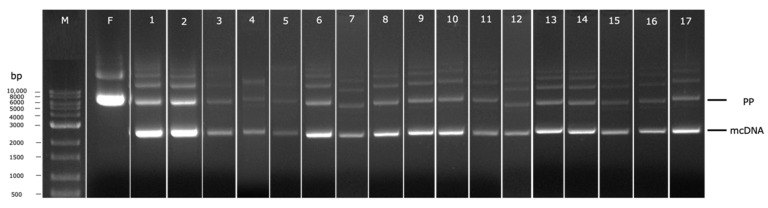
Analysis of PP recombination into mcDNA by agarose gel electrophoresis. M–molecular weight marker. F-Fermentation at 42 °C. Numbers 1–17 corresponds to each condition generated by DoE, as indicated in Table 1.

**Table 1 biomedicines-10-00990-t001:** Percentage of mcDNA recombined obtained for each condition of DoE tested.

Assay Number	Temperature (°C)	Antibiotic Concentration (µg/mL)	Induction Time (h)	mcDNA Recombined (%)
**1**	30	50	1	81.43
**2**	30	0	1	92.75
**3**	30	25	3	58.73
**4**	30	50	5	26.98
**5**	30	0	5	47.85
**6**	34	25	1	57.78
**7**	34	0	3	22.60
**8**	34	25	3	42.01
**9**	34	25	3	41.24
**10**	34	25	3	40.95
**11**	34	50	3	28.55
**12**	34	25	5	23.40
**13**	38	50	1	43.40
**14**	38	0	1	38.84
**15**	38	25	3	21.56
**16**	38	50	5	16.51
**17**	38	0	5	18.68

**Table 2 biomedicines-10-00990-t002:** Statistical coefficients of mcDNA optimization.

Output	R^2^	Adjusted R^2^	Predicted R^2^	Adequate Precision
**% of mcDNA**	0.9972	0.9937	0.9762	60.885

**Table 3 biomedicines-10-00990-t003:** ANOVA analysis for response surface quadratic model for the % of recombined mcDNA. *p*-value < 0.05 is considered significant.

Source	Sum of Squares	dF	Mean Square	F Value	*p* Value
**Model**	6526.52	9	725.17	280.41	<0.0001
**Temperature (A)**	2520.16	1	2520.16	974.50	<0.0001
**[Antibiotic] (B)**	129.17	1	129.17	49.95	0.002
**Induction time (C)**	3268.14	1	3268.14	1263.73	<0.0001
**AB**	149.47	1	149.47	57.80	0.0001
**AC**	341.91	1	341.91	132.21	<0.0001
**BC**	33.13	1	33.13	12.81	0.0090
**A^2^**	50.20	1	50.20	19.41	0.0031
**B^2^**	0.88	1	0.88	0.34	0.5776
**C^2^**	0.14	1	0.14	0.053	0.8244
**Residual**	18.10	1	2.59	−	−
**Lack of fit**	17.50	5	3.50	11.66	0.0808

**Table 4 biomedicines-10-00990-t004:** Predicted outputs for optimal point. CI–Confidence Interval.

Output	Predicted Mean	SE Mean	95% CI Low	95% CI High	SE Predicted	95% PI Low	95% PI High
**% of recombined mcDNA**	92.1131	1.43	88.72	95.50	2.15	87.02	97.21

**Table 5 biomedicines-10-00990-t005:** Determination of L-arabinose consumption during the bioreactor induction step.

Time (h)	Initial Concentration of L-Arabinose (%)	
	0.010	1.000
**2**	0.006 ± 0.0012	0.868 ± 0.0150
**4**	0.005 ± 0.0025	0.791 ± 0.0112
**6**	0.003 ± 0.0003	0.774 ± 0.0209
**8**	0.003 ± 0.0002	0.729 ± 0.0183
**10**	0.002 ± 0.0012	0.707 ± 0.0106

**Table 6 biomedicines-10-00990-t006:** Comparison of PP and mcDNA concentration in orbital shaker and bioreactor.

Nucleic Acid	Orbital Shaker	Increment	Bioreactor
**PP**	23.07 mg/L	65.46	1.51 g/L
**mcDNA**	16.48 mg/L	69.97	1.15 g/L

## Data Availability

Not applicable.

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
