# Peer review of "Maximization of the Minicircle DNA Vaccine Production Expressing SARS-CoV-2 RBD"

_biomedicines, 2022, doi:10.3390/biomedicines10050990_

Round 1

Reviewer 1 Report

The report by Ventura and co-workers deal with studies aimed at improved minicircle DNA to be used as vaccines. An experimental vaccine model against COVID-19 Is used. This is a relevant topic, due to the increasing importance of plasmid DNA in the manufacture of biopharmaceuticals (adenoviruses, DNA vaccines, mRNA vaccines, cell-free protein synthesis systems) and as a pharmaceutical active ingredient. In fact, the first DNA vaccine for use in humans was approved in India last year. It is a vaccine against COVID-19. The analytical methods are well explained. The experimental design seems appropriate.

To this reviewer´s opinion, there are some relevant issues that should be clarified.

  1. The authors decided to use complex media for their cultures. However, chemically defined media are preferred in industry. Please elaborate.
  2. The authors decided to use DOT> 30%. However, it has been reported that low DOT can favor plasmid content per cell (doi.org/10.1016/j.bej.2019.107303; Passarinha LA, Diogo MM, Queiroz JA, Monteiro GA, Fonseca LP, Prazeres DMF (2006) Production of ColE1 type plasmid by Escherichia coli DH5α cultured under nonselective conditions. J Microbiol Biotechnol 16:20–24)
  3. DOT = 60 % increased the amount of produced biomass. However, all the experiments were performed under non-limiting oxygen consumption. How can the authors explain this increase on biomass synthesis?
  4. Figure 7 is not necessary.

Otherwise, the report seems reasonable.

Author Response

Dear Reviewer 1:

The authors would like to acknowledge the careful evaluation and pertinent comments and the possibility to improve our manuscript. All the questions were answered in the present document and the recommended modifications were made, being properly highlighted at yellow in the revised manuscript file.

Reviewer 1:

The report by Ventura and co-workers deal with studies aimed at improved minicircle DNA to be used as vaccines. An experimental vaccine model against COVID-19 Is used. This is a relevant topic, due to the increasing importance of plasmid DNA in the manufacture of biopharmaceuticals (adenoviruses, DNA vaccines, mRNA vaccines, cell-free protein synthesis systems) and as a pharmaceutical active ingredient. In fact, the first DNA vaccine for use in humans was approved in India last year. It is a vaccine against COVID-19. The analytical methods are well explained. The experimental design seems appropriate. To this reviewer´s opinion, there are some relevant issues that should be clarified. Otherwise, the report seems reasonable.

Response: We deeply thank the Reviewer for this comment, which gives us the possibility to improve the Introduction section of the manuscript. Indeed, a DNA vaccine against COVID-19 was approved in India last year [1]. The ZyCoV-D was the world’s first plasmid DNA vaccine approved for human use, comprising the plasmid DNA vector pVAX1, carrying the gene encoding the S glycoprotein and the sequence encoding for the IgE signal peptide. This vaccine was the first COVID-19 vaccine approved for young adults older than 12 years. This information has been included in the revised version of the manuscript (please, see page 2). Additionally, the following reference has been included in the list of references of the revised manuscript.

  1. Hassine, I.H. Covid-19 vaccines and variants of concern: A review. Med. Virol. 2021, e2313.

  1. The authors decided to use complex media for their cultures. However, chemically defined media are preferred in industry. Please elaborate.

Response: We are grateful for the reviewer's comment, which we will try to address taking into account the strategy followed in the upstream stage design described in the presented work. First at shaking cultures scale and in order to identify the ideal growth conditions, three inputs such as the antibiotic concentration, the induction time and the induction temperature were established as vital, by the Design of Experiments (DoE), to optimize the PP recombination into mcDNA vector. These preliminary studies were conducted on the Terrific broth (TB – complex media for E. coli growth) and Luria Bertani (LB – complex media for induction). After the validation of the optimal point obtained by the DoE, the main inputs were subjected to a scale-up under a mini bioreactor platform. Therefore, it was imperative to keep the culture media used on orbital shakers at the bioreactor trials, in order to compare mcDNA yield levels obtained within the scale increment. It is well known that the formulation of the cultivation medium affects dramatically the performance and nature of E. coli growth and this can be limited by specific growth factors and other nutritional requirements including carbon, nitrogen, phosphorus, sulfur, magnesium and potassium iron, manganese, zinc, copper. Also, when the intention is to develop and identify a mathematical model that controls the specific rate of cell growth based on the type and concentrations of the main sources of carbon (glucose, glycerol, among others) and/or nitrogen (ammonium sulfate, amino acids, etc.), the application of defined or semi-defined is a fundamental requirement. We should reinforce that the modeling based on an exhaustive medium formulation was not the aim of this work. Nevertheless, our research group proved that the introduction of glucose in TB leads to a reduction in final cell growth and plasmid production (therapeutic vector pcDNA3–FLAG–p53). Additionally, an increase of glucose concentration to 10 g/L, limited even more, cell growth of the Escherichia coli VH33. Indeed, under a shaker scale, we observed that medium TB (without glucose) presents the best results in terms of plasmid DNA volumetric and specific yields and purity [2]. It is well known that an excess of glucose in the media can lead to growth inhibiting concentrations of acetate in different E. coli strains, blocking the expression of target biomolecules at considerably lower culture densities. Obviously, there are several strategies reported in the literature to reduce and cancel the effects of the presence of acetate and which may consider the use of glycerol instead of glucose. In this work, none of the media contains glucose. As can be consulted in Materials and Methods section, the TB medium was formulated with 4 mL/L of glycerol; in order to avoid inhibitory accumulations of acetate for cell growth. Here, we can consider the TB medium as a “semi-defined” medium since we know one of the main carbon sources. In the spite of the impact of changing the base formulation of the culture media has not yet been analyzed. However, we already have some indications based on production work with other vectors and data from this work supporting the design of a new defined (such as the M9 minimal medium) or semi-defined medium that could maintain or increase the outputs achieved here. Overall we will consider this approach in a future research work, in order to reduce the base costs of the upstream stage so that they can be easily transposed to the industrial scale.

  1. Martins, L.M.; Pedro, A.Q.; Oppolzer, D.; Sousa, F.; Queiroz, J.A.; Passarinha, L.A. Enhanced biosynthesis of plasmid DNA from Escherichia coli VH33 using Box–Behnken design associated to aromatic amino acids pathway. Eng. J. 2015, 98, 117-126.

  1. The authors decided to use DOT> 30%. However, it has been reported that low DOT can favor plasmid content per cell (doi.org/10.1016/j.bej.2019.107303; Passarinha LA, Diogo MM, Queiroz JA, Monteiro GA, Fonseca LP, Prazeres DMF (2006) Production of ColE1 type plasmid by Escherichia coli DH5α cultured under nonselective conditions. J Microbiol Biotechnol 16:20–24)

Response: We thank the reviewer for the pertinent comment. In this research work, we applied the pMC.CMV-MCS-EF1-GFP-SV40Poly A plasmid encoding the RBD of S protein from SARS-CoV-2, a larger vector with 7.7 kbp and high copy number per cell; in comparison to the lower size plasmids applied by our research group (Passarinha et al. “Production of ColE1 type plasmid by Escherichia coli DH5α cultured under nonselective condition”); pVAX1-LacZ with 6.1 kbp and pVAX1 - a 3 kbp high copy number plasmid applied by Lara et al. in “Effect of the oxygen transfer rate on oxygen-limited production of plasmid DNA by Escherichia coli”. Typically, the choice of the target plasmid and size may have a great influence on the recombination process during de fermentation stage. Thus, we took advantage of minicircle technology with the pMC.CMV-MCS-EF1α-GFP-SV40polyA, the parental plasmid cloning vector for episomal expression of DNA and transformed the ZYCY10P3S2T E. coli Minicircle Producer Strain. Additionally, plasmid-bearing cells exhibit an altered profile of central metabolic gene expression and a slower growth rate when compared to plasmid-free cells. As described by Passarinha and co-workers, a decrease in the DOT from 60% to 5% was shown to increase the specific pDNA (pVAX1-LacZ, 6.1 kb) concentration approximately 1.5-fold and led to a 2.2-fold increase in the purity of cell lysates. However, the use of higher DOT led to 2.8-fold higher volumetric plasmid productivity as a consequence of a faster growth rate, reducing the fermentation time from 24 to 8 h. Also, Lara and collaborators demonstrated that the pDNA supercoiled fraction (3 kbp) was greatest in cultures at an OTRmax of 30 mmol L−1 h−1 (higher DOT), reaching 92.9% for the wild type strain and 98.7% for the VHb-expressing strain. So it will be sustainable to use DOTs greater than 30% to promote the expression of plasmids with a higher molecular weight, such as the pMC.CMV-MCS-EF1-GFP-SV40Poly A vector. Indeed, the best conditions in the mini-bioreactor platform scale-up were applying 60% pO2 in the fermentation step during 5 h and 30% pO2 in the induction step. The difference between the two percentages of dissolved oxygen between the cell growth phase and the induction phase, ensured that structural and segregation stability was not affected mainly in the induction step, giving rise to an effective recombination process as demonstrated by the data obtained. The yield of mcDNA-RBD was increased to a concentration of 1.15 g/L, when compared to the orbital shaker studies (16.48 mg/L). In addition, the increase of cellular metabolism can interfere with the time that cells need for the expression of ΦC31 integrase and endonuclease I-SceI and their action into the PP recombination and miniplasmid degradation [3]. This information was clarified in the manuscript (please, see page 19).

  1. Carnes, A.E.; Hodgson, C.P.; Williams, J.A. Inducible Escherichia coli fermentation for increased plasmid DNA production. Biotechnol. Appl. Biochem. 2006, 45, 155.

  1. DOT = 60 % increased the amount of produced biomass. However, all the experiments were performed under non-limiting oxygen consumption. How can the authors explain this increase on biomass synthesis?

Response:

We appreciate the reviewer's comments. In fact, experiments were carried out under oxygen-limiting conditions, guaranteeing basal levels of oxygen in the bioreactor during the induction phase (results not shown). The strategy was based on the assumption that the removal of oxygen under the induction step will "force" the E. coli cells to capture more L-arabinose, favoring PP recombination into mcDNA. The conditions were as follows: in the fermentation step the pressure used was 60%, in which we had already verified that there was more amplification of our PP, with 65.46x greater than shaker flasks (1.51 g/L). Then we applied a DOT of 0% in the induction stage, which allowed to achieve 58.19x more mcDNA than in shaker flasks (0.958 g/L). Nevertheless, our optimal condition for induction implied the use of a DOT of near 30% leading to 69.97x more mcDNA (1.15 g/L). To point out that the optical density remained between 30-34 during the 10 h of the trial. In conclusion, the increase in biomass during the growth phase may be related to the synergy achieved between a DOT of 60% and the use of TB medium, rich in sources such as yeast extract and tryptone, that favor the specific rate of cell growth maintaining the native plasmid integrity. Something that is not observed when we reduced DOT to 30% during the initial phase of fermentation.

  1. Figure 7 is not necessary.

Response: Considering the reviewer suggestion, the Figure 7 was removed from the revised manuscript.

Reviewer 2 Report

The work by Ventura et al demonstrate the generation of a mini circle DNA expression vector that encodes SARS-CoV-2 receptor biding domain (RBD). They establish a DoE to optimize mcDNA vector generation from the parental plasmid and provide an optimization protocol best suited for mcDNA biosynthesis improvement and high throughput scalability. Their aims included generation of the mcDNA-RBD vector as well as improving overall mcDNA yields by establishing set parameters for culturing and production.

Major Comments:

The work overall is very interesting and provides detailed descriptions from generation of mcDNA vector to production. I would suggest that the authors modify the title to “Maximization of the minicircle DNA vaccine production expressing SARS-CoV-2 RBD” The current title may suggest the construct was used as a candidate against COVID-19, however this manuscript highlights the production not downstream immunogenicity and protective efficacy testing against SARS-CoV-2.

The impact of the manuscript can also be increased if the authors demonstrate that their mcDNA-RBD vector expresses in vitro using western blots.

Figure 3. Can the authors include a gel picture of the final mcDNA-RBD construct showing the vector and RBD insert is present? This figure shows each individual digest either vector alone or RBD insert alone.

Figure 4. Please indicate which sequence belongs to RBD and which is RBD vector clone. Please include details about what sequence was used for this alignment including NCBI accession number. An amino acid alignment would also be helpful to include in this figure.

Figure 5. Please include a molecular marker for this gel image. OD of mcDNA after each experimental condition would demonstrate increasing total mcDNA concentrations with the addition of increased temperature and induction with centrifugation.

Figure 6. Please include molecular marker. Please move the mcDNA arrow so it does not block the gel image. Were all the samples run independently or is there a gel which shows all the bands? This would make it easier to observe size differences due to technical running issues or other gel run artifacts.

In line 52 please include both innate and adaptive immune response since both have been implicated in SARS-CoV-2 pathogenesis.

In line 70 to 72 the authors say …the presence of those sequences in the DNA vector delivered to a patient can potentially trigger adverse immune response and the antibiotic resistance gene can be potentially transferred into human microflora..

Please discuss the potential caveat to this in parental plasmid DNA contaminated mcDNA preps.

Line 118 please detail where the sequence originated, was this an original parental Wuhan strain RBD? Can the authors provide and NCBI accession number?

Author Response

Dear Reviewer 2:

The authors would like to acknowledge the careful evaluation and pertinent comments and the possibility to improve our manuscript. All the questions were answered in the present document and the recommended modifications were made, being properly highlighted at yellow in the revised manuscript file.

Reviewer 2:

The work by Ventura et al demonstrate the generation of a minicircle DNA expression vector that encodes SARS-CoV-2 receptor biding domain (RBD). They establish a DoE to optimize mcDNA vector generation from the parental plasmid and provide an optimization protocol best suited for mcDNA biosynthesis improvement and high throughput scalability. Their aims included generation of the mcDNA-RBD vector as well as improving overall mcDNA yields by establishing set parameters for culturing and production. 

Major Comments:

The work overall is very interesting and provides detailed descriptions from generation of mcDNA vector to production. I would suggest that the authors modify the title to “Maximization of the minicircle DNA vaccine production expressing SARS-CoV-2 RBD” The current title may suggest the construct was used as a candidate against COVID-19, however this manuscript highlights the production not downstream immunogenicity and protective efficacy testing against SARS-CoV-2.

Response: We deeply thank the Reviewer for this pertinent comment, and we strongly agree that the current title might suggest that minicircle DNA vaccine was already used as a candidate against COVID-19. The title of the manuscript was changed to “Maximization of the minicircle DNA vaccine production expressing SARS-CoV-2 RBD” in the revised manuscript file.

The impact of the manuscript can also be increased if the authors demonstrate that their mcDNA-RBD vector expresses in vitro using western blots.

Response: We thank the Reviewer for this comment, and we agree that western blot analysis would bring relevant information on the mcDNA-RBD vector expression. However, in the present work, the main aim was to maximize the minicircle DNA vaccine production encoding the RBD gene. In fact, we are working on the formulation of different nanosystems based on polyethylenimine (PEI), cholesterol and mannose for delivering this mcDNA-RBD vector to antigen-presenting cells (APCs), and these results will be included in a future publication. At this moment, we have already confirmed by RT-PCR that the mcDNA-RBD vector reaches the nucleus of APCs and the gene of interest is transcribed into the respective messenger RNA (detected with specific primers), as demonstrated in Figure 1.  

Figure 1: Evaluation of RT-PCR products by agarose gel electrophoresis for RBD transcripts amplification in JAWSII cells transfected for 24 h. M - Molecular weight marker; (-) - Negative control; CT - Non-transfected cells. Cells were transfected with: Lane 1 – PEI/PP-RBD; Lane 2 – PEI/mcDNA-RBD; Lane 3 - PEI-cholesterol/PP-RBD; Lane 4 - PEI-cholesterol/mcDNA-RBD; Lane 5 - PEI-cholesterol-mannose/PP-RBD; Lane 6 - PEI-cholesterol-mannose/mcDNA-RBD.

Thus, RBD transcripts detection revealed that all nanosystems were able to induce the expression of the target antigen, although PEI-cholesterol/mcDNA-RBD and PEI-cholesterol-mannose/mcDNA-RBD performance stood out. Furthermore, we will start the RBD transcripts assessment by real-time PCR and we are in the process of acquiring the necessary antibodies to carry out western blot studies. Taking this into account, and unfortunately, it is not possible to perform these experiments and report the respective results to respond on time, considering the proposed deadline for the revision. We will include those results in a future manuscript on the development of multifunctional nanoparticles for targeted delivery of minicircle DNA vaccine against COVID-19 to antigen-presenting cells.

Figure 3. Can the authors include a gel picture of the final mcDNA-RBD construct showing the vector and RBD insert is present? This figure shows each individual digest either vector alone or RBD insert alone.

Response: We thank the Reviewer for this comment and the possibility to improve our manuscript regarding the cloning of the RBD gene sequence into the PP vector. Indeed, Figure 3 shows the initial PP vector and the further digestions performed with XbaI and BamHI enzymes both in the PP vector and RBD insert. Considering the reviewer's suggestion, Figure 3 was reorganized to include the initial PP vector and the final PP-RBD vector (which due to the presence of RBD sequence (640 bp) slightly increase its size when compared to PP vector). This gel picture has been included in the revised version of the manuscript (please, see Figure 3).

Figure 4. Please indicate which sequence belongs to RBD and which is RBD vector clone. Please include details about what sequence was used for this alignment including NCBI accession number. An amino acid alignment would also be helpful to include in this figure.

Response: We thank the Reviewer for this comment and the possibility of clarifying some aspects related to the cloning of RBD gene into the PP vector. The Subject is the sequence corresponding to RBD of S protein from SARS-CoV-2 and Query corresponds to our PP-RBD vector sequence after sequencing, provided by STAB VIDA company (Caparica, Portugal). For the alignment, we selected the S surface glycoprotein [ Severe acute respiratory syndrome coronavirus 2 ] with the sequence: NC_045512.2 (21563..25384). However, we performed the alignment only with the sequence corresponding to the secreted receptor-binding domain (RBD; a.a. 333-529) of S protein from SARS-CoV-2, as indicated in the template plasmid used for the RBD amplification (Addgene Plasmid # 145145). For analyzing these 197 amino acids, we used the Protein ID: YP_009724390.1, corresponding to spike glycoprotein. Then, we analyzed the corresponding sequence in the Addgene Plasmid (http://www.addgene.org/browse/sequence/324821/) and performed the nucleotide alignment with our PP-RBD vector. As suggested by the Reviewer, an amino acid alignemt was also included in this Figure. Furthermore, the origin of the Subject and Query sequences and the NCBI accession number have been included in the revised version of the manuscript (please, see Figure 4).

Figure 5. Please include a molecular marker for this gel image. OD of mcDNA after each experimental condition would demonstrate increasing total mcDNA concentrations with the addition of increased temperature and induction with centrifugation.

 Response: We deeply thank the reviewer for this comment and the possibility to improve our manuscript. In fact, we measured the OD (260 nm) of each sample at the end of the extractions with the miniprep kit and correspond to the following DNA total concentrations: lane 3 - 60 mg/L; lane 4 - 64 mg/L; lane 5 - 73 mg/L; lane 76 mg/L. Thus, it is visible an increase of the mcDNA content through the electrophoresis as a function of the increased temperature and induction with centrifugation, which is also reflected in the slight increase of the total DNA concentration. This information was included in the revised version of the manuscript, in the Materials and Methods section, topic “2.3. Fermentation and induction conditions in an orbital shaker” (please, see page 5), in the Results section, topic “3.2. PP amplification and recombination into mcDNA” (please, see page 11) and the gel image has been updated (please, see Figure 5).

Figure 6. Please include molecular marker. Please move the mcDNA arrow so it does not block the gel image. Were all the samples run independently or is there a gel which shows all the bands? This would make it easier to observe size differences due to technical running issues or other gel run artifacts.

Response: We deeply appreciate the comment made by the reviewer, which allowed us to turn our results more perceptible and clearer. The samples are all separated because they were organized following the sequence of experiments suggested by the DoE, since it is more perceptible the visualization of the conditions proposed by the computational program. In fact, some samples were run simultaneously on the same gel because we considered the DoE proposed experiments in subgroups to conduct at the same time the experiments with similar production conditions, specifically concerning the same temperature. At the end of production assays of each subgroup, we directly follow with the collection of a homogeneous sample of cells, with the same volume of culture by each conducted assay, to perform the respective DNA extraction by the miniprep kit, OD assessment and electrophoresis visualization. The direct treatment of samples subgroups is preferred to minimize as much as possible the effects/artifacts that the samples could suffer, such as the loss or conversion of the PP or mcDNA supercoiled isoforms into open circular isoforms or degradation that can occur through the waiting times to perform electrophoresis of all samples together. Even with the freezing and thawing of samples obtained during the first tested conditions, it is not possible to perform all the production conditions under different temperatures on the same day. In addition, we also tried to standardize as much as possible the visualization of the samples by agarose electrophoresis, by using the same conditions in the preparation of electrophoresis gels, run time and UV intensity during the visualization of electrophoresis gels, to avoid any external influence on the evaluated samples. This information has been included in the revised version of the manuscript, in the Materials and Methods section, topic “2.5. Agarose gel electrophoresis” (please, see page 6) and the gel image has been updated (please, see Figure 6).

In line 52 please include both innate and adaptive immune response since both have been implicated in SARS-CoV-2 pathogenesis.

Response: We thank the Reviewer for the comment and the possibility of further improving the manuscript. This information has been included in the Introduction section of the manuscript (please, see Page 2).

In line 70 to 72 the authors say … the presence of those sequences in the DNA vector delivered to a patient can potentially trigger adverse immune response and the antibiotic resistance gene can be potentially transferred into human microflora ... Please discuss the potential caveat to this in parental plasmid DNA contaminated mcDNA preps.

Response: We sincerely thank the Reviewer for this comment, which allow us to improve the Introduction section of the manuscript. There is a small amount of PP that may be present at the end of the recombination process, however, the adverse responses will be reduced when compared to vaccines or therapies based merely on plasmid DNA. Furthermore, our work aims to minimize the presence of PP that has not been recombined. In addition, after the fermentation and recombination process, the samples obtained can be purified with an adequate method to eliminate the residual PP with techniques that have already been studied by our research group, such as molecular exclusion chromatography with Sephacryl S-1000 SF and with cadaverine modified monoliths. Both methods presented a good selectivity to mcDNA, allowing successful isolation of supercoiled mcDNA from Escherichia coli lysate samples [4,5]. This information has been included in the Introduction section of the manuscript (please, see Page 2).

  1. Almeida, A.M.; Eusébio, D.; Queiroz, J.A.; Sousa, F.; Sousa, A. The use of size-exclusion chromatography in the isolation of supercoiled minicircle DNA from Escherichia coli lysate. J. Chromatogr. A. 2020, 1609, 460444.
  2. Almeida, A.M.; Černigoj, U.; Queiroz, J.A.; Sousa, F.; Sousa, A. Quality assessment of supercoiled minicircle DNA by cadaverine-modified analytical chromatographic monolith. J. Pharm. Biomed. Anal. 2020, 180, 113037.

Line 118 please detail where the sequence originated, was this an original parental Wuhan strain RBD? Can the authors provide and NCBI accession number?

Response: We thank the Reviewer for this comment, which gives us the possibility to clarify the manuscript. Indeed, the RBD sequence is from the original parental Wuhan strain. The NCBI Reference Sequence: NC_045512.2, corresponding to the [ Severe acute respiratory syndrome coronavirus 2 isolate Wuhan-Hu-1, complete genome ] was used by Dr. Erik Procko and co-workers to construct the pcDNA3-SARS-CoV-2-S-RBD-8his plasmid (Addgene - Plasmid #145145). Then, the S surface glycoprotein [ Severe acute respiratory syndrome coronavirus 2 ] with the Gene ID: 43740568 and Sequence: NC_045512.2 (21563..25384) was considered, which the amino acids 333-529 correspond to the secreted receptor-binding domain (RBD). This information has been included in the revised version of the manuscript (please, see Page 4).

Round 2

Reviewer 1 Report

I can now recommend to accept the manuscript.

Reviewer 2 Report

The authors have addressed all my comments and have responded appropriately.